# Cybervictimisation and Well-Being during the Outbreak of COVID-19: The Mediating Role of Depression

**DOI:** 10.3390/healthcare10091627

**Published:** 2022-08-26

**Authors:** Anna Lisa Palermiti, Maria Giuseppina Bartolo, Rocco Servidio, Angela Costabile

**Affiliations:** Department of Cultures, Education and Society, University of Calabria, 87036 Arcavacata di Rende, Italy

**Keywords:** COVID-19, cybervictimisation, depression, well-being

## Abstract

Social distancing during the COVID-19 pandemic contributed to modifying relational habits and increasing Internet use to engage in antisocial behaviours such as cybervictimisation. Additionally, social distancing can reinforce the relationship with internalising behaviours such as depression. Through an adolescent sample, this study examines the relationship between cybervictimisation and well-being and the mediating role of depression. The hypothesis was tested via Structural Equation Model (SEM) analysis to verify the role of depression as a mediator between cybervictimisation and well-being. The main results reveal that the effect of cybervictimisation on well-being was fully mediated by depression. The findings should stimulate debate on possible interventions to promote adolescent well-being and to avoid emotional and mental health problems related to social isolation.

## 1. Introduction

Over the last two years, the COVID-19 pandemic has affected people worldwide not only in terms of their health but also on a psychological level. During the pandemic, one of the most significant forms of intervention to reduce the transmission of the virus has been the closing of schools and home confinement. However, the COVID-19 pandemic left many children and teens feeling lonely: often, their only social activities and relationships were online. Being isolated from friends, educators, colleagues, and mentors made them more prone to losing confidence and motivation. In addition, these social situations and factors may have led adolescents to become perpetrators or victims of cyberbullying [1], as during the COVID-19 pandemic, people spent more time online and used Internet technologies, such as social media applications, to communicate with others [2]. Combined with social isolation, these aspects may bring a range of psychological harms [3] and may negatively affect the physical and mental health and well-being of children [4,5].

Cybervictimisation—that is, victimisation experiences that occur through digital media—is a widely studied phenomenon [6,7,8]. It is defined as any aggressive online behaviour that inflicts harm or discomfort on victims through aggressive messages or acts via digital devices with Internet access [9,10,11]. This phenomenon is characterised by asynchronicity and anonymity, which stimulate disinhibited behaviours and conceal the identities of cyberbullies from their victims through tools such as e-mails, texting, instant messaging [12,13], and social networking (e.g., Facebook, Twitter, YouTube) [14,15].

Cybervictimisation attracts a great deal of interest from educators, parents, researchers, and the general public, as it correlates with social functioning, behavioural problems, and psychological health [16,17,18]. Indeed, different studies evidence that cybervictimisation is significantly associated with loneliness or social isolation, negative self-cognition [19,20,21], negative social comparison [22], low self-esteem [23,24], hopelessness [25], maladaptive emotion regulation [22], sleeping difficulties [26], and distress [27], leading to serious consequences, particularly for the victims, including depression and anxiety [19,28,29,30]. 

Prior cross-sectional and longitudinal studies have examined the relationship between cybervictimisation and depression, indicating that cyberbullying positively predicts depressive symptoms [31,32]. More specifically, a growing number of cross-sectional and longitudinal studies show that cybervictimisation is significantly associated with depression [28,30,33]. The consequences of cybervictimisation could be related to the specific characteristics of the phenomenon, such as anonymity and the rapid spread of information on the web [16], which lead cybervictims to experience feelings of sadness, emptiness, and/or irritability or to display avoidance behaviours. In addition, the possibility of receiving threatening or hurtful messages anywhere and at any time of the day through their electronic devices [34] makes victims feel that the consequences of these acts are irreparable, thereby increasing their depression level. 

While numerous studies have identified a link between cybervictims and high levels of depression [35,36], fewer investigations have explored the relationship between cybervictimisation and low levels of subjective well-being [29,37,38]; more research is therefore required.

It is widely known that well-being is a subjective state of happiness [39]. It involves personal life satisfaction and positive relationships [40], which is crucial for teens trying to build their identity. Interest in the study of well-being has been growing over the last few years, as researchers have investigated the quality of life of people [41]. Well-being is a protective factor against internalising symptoms such as depression and psychosomatic problems [42]. Unfortunately, cybervictimisation experiences can negatively impact psychological health, safety, and well-being [38], particularly among adolescents experiencing physical, emotional, and social changes that can lead to stress, confusion, and emotional instability [43]. Some researchers have posited that cybervictimisation has more destructive effects than traditional bullying [25,44] for reasons such as the anonymity of the bully and the lack of supervision [45]. Considering the severity of the effects of cybervictimisation on adolescents, we hypothesised a negative relationship between cybervictimisation and well-being. Being subjected to digital threats, insults, and denigration puts the victim in a state of malaise that prevents him or her from peacefully experiencing any kind of social relationship, whether online or offline, since it causes unhappiness. 

Some behaviours, such as cybervictimisation [46], seem to increase in the presence of stressful events. Stressful life events are a broad construct that incorporates adverse social–environment experiences involving various domains, including mental states and social relationships [47]. Stressful life events are reported to be prevalent among adolescents [48] since adolescence is a life phase marked by numerous physiological (e.g., sexual maturity), psychosocial (e.g., self-identity, independence), and social–environmental (e.g., relationships, study environment) changes [49,50]. These changes make adolescents more vulnerable to depression after stressful experiences [51]. The COVID-19 pandemic can be considered a highly stressful event in the lives of all people, and especially for adolescents, due to lockdowns, social isolation, and the exclusively online pursuit of study and interpersonal social activities. We can therefore hypothesise that the COVID-19 pandemic had a negative impact when it comes to cybervictimisation. At the same time, considering that cybervictimisation is an adverse experience that can decrease well-being and that depression can play a crucial role in the relationship between cybervictimisation and well-being [16,52], it may be inferred that—as already noted above—depression can impair well-being in adolescents. In this sense, it is reasonable to assume that depression can mediate the relationship between cybervictimisation and adolescent well-being. Indeed, a previous study highlighted the negative influence of cybervictimisation on well-being [38] and a significant association between cybervictimisation and depression in adolescence [53]. 

Based on the literature mentioned above, the present study differs from the previous ones as it investigates the relationship between cybervictimisation and well-being and the mediating role of depression. In addition, the current study tries to identify risk factors to be considered in the design of intervention programmes to prevent cybervictimisation and increase well-being.

Hence, the current study investigates the relationship between cybervictimisation and well-being among adolescents in Italy. We hypothesised a negative association between cybervictimisation and well-being and that depression may mediate this association. Moreover, we hypothesised that COVID-19 is a stressful event that can affect the lives of adolescents by having adverse effects on cybervictimisation and depression. 

## 2. Materials and Methods

### 2.1. Participants and Procedures

The questionnaire was collected via an online survey system in April–May 2020, (LimeSurvey). We recruited a convenience sample of 711 Italian students by using a snowball sampling technique, where the first participants were recruited among friends and acquaintances who publicised the questionnaire and recruited other participants among their classmates, friends, or relatives. The two inclusion criteria were the age range (15–25) and the student status. For an acceptable proportion of participants (*N* = 131, 18.43%), information with regard to one or more study variables was missing. Therefore, these participants were excluded from the analyses. Thus, the final sample consisted of 580 adolescents: 83.1% (482) of the sample were females. Their ages ranged from 15 to 25, with a mean age of 19.99 years (SD = 2.72).

All the study procedures and materials were designed and employed according to the ethical standards laid out by the Italian Psychological Association (AIP). All participants provided informed consent, as did the parents of minors. All participants were ensured the anonymity and confidentiality of their answers. They were provided with information about the nature and purpose of the study and were also informed about their right to stop at any time without having to provide any justification. It took participants about 20 min to complete the anonymous online survey.

### 2.2. Measures

The participants completed a battery of validated self-report measures and one section to collect demographic information (i.e., gender, age, educational level).

-Depression Anxiety Stress Scales-21 (DASS-21)

Negative emotional states were assessed using the Italian version of DASS-21 [54]. This is a self-report instrument consisting of three 7-item subscales designed to assess the level of depression, anxiety, and stress of a person over the last week. Item examples include “I felt that I had nothing to look forward to”. Only the subscale for depression (α = 0.90) was used for the current study. The responses are given on a 4-point Likert scale, ranging from 0 (does not apply to me at all) to 3 (applies to me most of the time), with higher scores indicating a more negative experience in the past week. Although the use of DASS-21 has sparked some debate in the literature concerning its applicability to adolescents, a previous study on cyberbullying applied this scale [55].

-Florence Cyberbullying–Cybervictimisation Scales (FCBVSs)

The FCBVSs [56] were used to assess cyberperpetration and cybervictimisation behaviours over the course of the two previous months. Each scale contains 14 items. One previous research supported the use of the FCBVSs as a second-order measure to obtain global scores for cyberbullying and cybervictimisation [56]. Items were scored on a Likert-type scale ranging from 1 (never) to 5 (several times a week). A mean composite score was calculated for each dimension of the scale. In accordance with Palladino and colleagues [57], the scales were introduced with the following sentence: “Cyberbullying is a new form of bullying, which involves the use of text messages, photos and videos, phone calls and e-mails to attack another student” (p. 113). For the current study, only the cybervictimisation subscale was used (α = 0.93).

-Warwick–Edinburgh Mental Well-Being (WEMWB)

The Italian version of WEMWB is a scale of 12 items (e.g., “I have been feeling optimistic about the future”), which are all positively worded [58]. In relation to each statement, respondents are required to describe their experience over the past two weeks using a 5-point Likert-type scale ranging from 1 (never) to 5 (always). A higher WEMWB score, therefore, indicates a higher level of mental well-being. The reliability value for the present study was excellent, α = 0.91.

-Fear of COVID-19 Scale (FCV-19S)

The Italian version of the FCV-19S [59] is a 7-item scale that assesses the fear of COVID-19. The seven items (e.g., “I am most afraid of coronavirus-19”) are rated on a 5-point Likert-type scale from 1 (strongly disagree) to 5 (strongly agree), with scores ranging from 7 to 35. The higher the score, the greater the fear of COVID-19. The scale showed an excellent value of reliability, α = 0.86.

### 2.3. Statistical Analyses

Preliminary data analyses were run by using the IBM SPSS Statistic program, version 26. Firstly, we computed descriptive statistics and tests to check the normality of the data. We had no missing data as we removed all the incomplete answers. Given that cyberbullying scales were not normally distributed, we transformed the scores with the support of SPSS by applying the van der Waerden ranking procedure [60] and used the transformed variables in all subsequent analyses. This procedure is useful to normalise the distribution of the data. Then, we computed bivariate correlations (Pearson’s r) among the variables of interest. Differences between the means of the variables were examined by computing a t-test for independent sample. Finally, before conducting the other analyses to test the hypotheses of the study, the reliability of the scales and subscales was estimated by computing Cronbach’s α.

Moreover, to test the hypotheses of the study, a structural equation modelling (SEM) analysis, using Mplus 7.01, was performed. The models were estimated with the maximum-likelihood parameter with standard errors and a mean-adjusted chi-square test statistic that was robust to non-normality (MLM). The MLM chi-square test statistic is also referred to as the Satorra–Bentler (S-B) chi-square. Gender and Fear of COVID-19 were also controlled. We assessed the fit of the tested models using the following multiple indexes: (a) comparative fit index (CFI) ≥ 0.95, (b) Tucker–Lewis Index (TLI) ≥ 0.95, (c) root mean square error of approximation (RMSEA) ≤ 0.06, and (d) standardized root mean square residual (SRMR) < 0.08 [61].

## 3. Results

### 3.1. Preliminary Analyses

The descriptive statistics (means and standard deviations for all the variables) and bivariate correlations between the variables are shown in Table 1.

Gender differences were found regarding cybervictimisation. The results reveal that males (M = 5.42, SD = 10.23) were significantly affected by cybervictimisation, t(578) = 7.72, *p* < 0.001, d = 0.57. There were no significant differences between gender, well-being, and depression.

### 3.2. Mediational Analysis

The hypothesised research model was tested with cybervictimisation as an independent variable, depression as a mediator, and well-being as the outcome variable, while the effect of gender and fear of COVID-19 were controlled. The results of the analysis indicate that the model fit well with the data: robust χ^2^ (196, *N* = 580) = 380.33, *p* < 0.001, CFI = 0.936, TLI = 0.925, RMSEA = 0.052, 90% CI [0.044, 0.060], SRMR = 0.049.

As shown in Figure 1, cybervictimisation was positively associated with depression, β = 0.195, *p* < 0.01. In turn, depression was negatively related to well-being, β = −0.631, *p* < 0.001. However, the direct effect of cybervictimisation on well-being was not significant (*p* > 0.05), indicating that depression fully mediated the hypothesised relationships. Specifically, cybervictimisation highlighted a robust total effect on well-being, β = −0.194, *p* < 0.01, with the mediation of depression (indirect effect), β = −0.123, *p* < 0.01. In addition, gender as a control variable affected cybervictimisation, β = −0.246, *p* < 0.001, with males being more at risk. Fear of COVID-19 instead positively influenced both cybervictimisation, β = 0.124, *p* < 0.05 and depression, β = 0.196, *p* < 0.001.

## 4. Discussion

The main findings of this study provide evidence that Italian adolescents with high levels of cybervictimisation show lower levels of well-being through the fully mediating role of depression and that the condition of cybervictimisation is not negatively associated with well-being directly. Our findings show a positive association between cybervictimisation and depression, as supported by a systematic review [62] that reports a strong association between cyberbullying and internalising symptoms. Some studies [23,34,63] affirm that being a victim is a stressful life condition for adolescents. Victims are afraid of experiencing harassment, threats, and negative comments every day, at any time of the day; in addition, they do not know who their attackers are, and this condition increases feelings of helplessness that, in turn, increase emotional distress, which over time may lead to depression. Indeed, youths who feel well and are satisfied with themselves and their lives are at lower risk of being victimised. Although some studies [28] underline that cybervictimisation experiences are similar in adolescents and emerging adults, insofar as they lead to the same outcomes (e.g., anxiety, depression, etc.), the results of the correlational analysis show a significant negative association.

In this study, we also considered gender differences. The results of the SEM analysis revealed that males were more likely to become cybervictims, as also pointed out by Wong and colleagues [64]. This association may be because males generally engage in online activities more—mainly contacting other people—which could lead to greater exposure to acts of cybervictimisation. However, some studies [29,37,38] have found a negative correlation between cybervictimisation and well-being. Furthermore, based on ecological frameworks [65], we have to consider that this study was conducted during the COVID-19 pandemic lockdown, which is to say during a specific time in which a stop had been put to most social activities. This condition of home confinement forced youths to spend more time on the Internet not only for study purposes but also to have fun or keep in touch with peers, thereby increasing their risk of falling into the cyberbullying trap [66].

The COVID-19 pandemic has had different psychological effects on adolescents and youths. Different studies conducted during highly traumatic situations such as pandemics or disasters have found high post-traumatic stress levels associated with depression, anxiety, and mental health problems in adolescents, especially young people who had been quarantined [67,68,69].

## 5. Limitations and Strengths

This study may be seen to present certain limitations.

Firstly, considering the situation, we had to use a convenience sample which limited generalizability to the broader population of youths. Secondly, we only investigated adolescents’ perceptions, so there is a lack of parental perspectives regarding family environments, and future studies should be developed in this direction. Thirdly, social pressure provides desirable responses, even in anonymous online questionnaires. Finally, a longitudinal study should examine the direction and the changing of some variables in light of the end of the emergency lockdown.

However, this study helps us assess cybervictimisation and mental health outcomes in Italy during the final weeks of the COVID-19 pandemic lockdown. It sheds light on emergency social conditions because it broadens our understanding of the relationships, social contexts, and aggressive online behaviour of adolescents. Furthermore, our findings enhance the knowledge of factors that can contribute to preventing cyberbullying and adverse conduct in adolescence.

The current results highlight that cybervictimization can put the well-being of young people and public health at risk, particularly considering the role of cyberbullying as a contributor to poor mental health and, potentially, suicidal ideation [4,5].

## 6. Conclusions

The findings of this study should stimulate reflection on the effects of living in social isolation. Our study suggests the need for immediate interventions to promote adolescent well-being and prevent severe behavioural, emotional, and mental health problems linked to the pandemic and social isolation.

Adolescents are the most vulnerable people and require careful consideration, so it is necessary:-to provide helpful information and support to adults on how this kind of stressful situation can be managed—for example, by talking to others about one’s fears, negative feelings, and emotions [70]. In this way, it is possible to help even the least resilient and most stressed parents to find ways to understand and support their children [71].-to develop support and prevention programmes for cybervictims that consider other variables, such as peer friendships, the family context, and support and coping strategies. As adolescents spent all their time at home with their families, we believe that reasonable parental control may have been able to mitigate the direct effects of cybervictimisation on well-being, acting as a resource against the negative impact of cybervictimisation [72,73,74].

In light of this, anticyberbullying intervention programmes should promote adolescent well-being to prevent health and social problems. Hellfeldt and colleagues [75] found that social support is a protective factor mediating cyberbullying and psychological well-being. Therefore, social support from one’s family and teachers appears to reduce the likelihood of depressive symptoms and anxiety and to increase well-being among young people. These findings should lead educators and health professionals to focus on the emotional well-being of adolescents.

## Figures and Tables

**Figure 1 healthcare-10-01627-f001:**
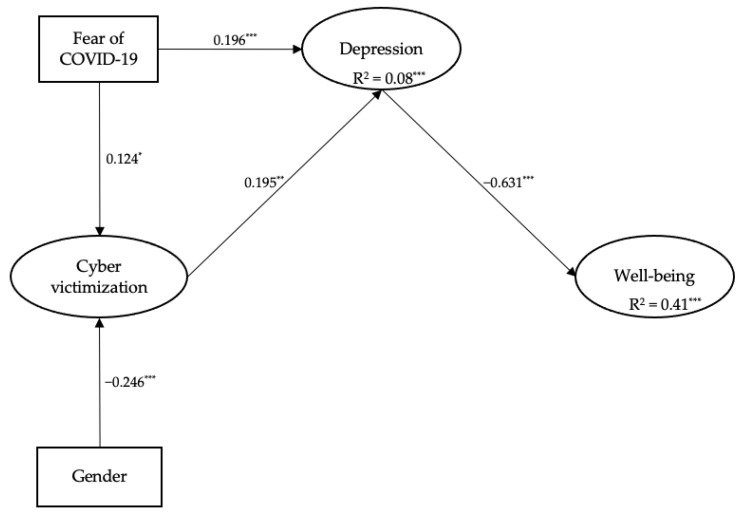
The standardised path coefficients of the mediational model. For clarity, only the significant relationships are depicted in the figure. Latent factors are presented in the circle, and measured variables are presented in the rectangles. Item factor loadings were all significant at *p* < 0.001. All the analyses were controlled for Fear of COVID-19 and gender (1 = Male, 2 = Female). * *p* < 0.05. ** *p* < 0.01. *** *p* < 0.001.

**Table 1 healthcare-10-01627-t001:** Mean, standard deviation, and Pearson bivariate correlations among study variables (*N* = 580).

	*M*	*SD*	1	2	3	4	5	6
Fear of COVID-19	2.51	0.91	-					
2. Cybervictimisation	1.82	5.32	0.05	-				
3. Well-being	38.43	8.85	−0.13 ***	−0.12 ***	-			
4. Depression	2.03	0.74	0.20 ***	0.17 ***	−0.58 ***	-		
5. Age	19.99	2.72	0.22 ***	−0.15 ***	−0.03	0.01	-	
6. Gender ^a^	-	-	0.23 ***	−0.23 ***	−0.02	0.03	0.39 ***	-

Note: *** *p* < 0.001. ^a^ Gender (1 = male and 2 = female) is a point-biserial correlation.

## Data Availability

The data presented in this study are available on request from the correspond author.

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
