# Peer review of "Cybervictimisation and Well-Being during the Outbreak of COVID-19: The Mediating Role of Depression"

_healthcare, 2022, doi:10.3390/healthcare10091627_

Round 1
Reviewer 1 Report
This is a clear, succinct, and well-written article. An important quality of this work is that it is accessible to readers who do not have a background in research methodology/statistics. It is not a surprise that depression is related to cyber-bullying and this article successfully promotes further research attention to adolescent cyber-bullying victimization and its consequences.
A strength of the article is the clear definitions of terms including cybervictimization, well-being, and stressful life events. With 77 references, the literature review is strong.
A sample size of 580 is very good, although I'm sure we'd all prefer for more than 17% of the sample to be male. The authors can make a higher percentage of adolescent males a priority in their follow-up research since males are more likely to be victims of cyberbullying.
While the placement of the paragraphs in the sections of 4. Discussion and 5. Limitations and strengths is okay, I'm going to suggest a bit of a reordering for clarity. When I am skimming a research article, after I look at the abstract, and the introduction, I look to the conclusion--I'm especially looking for specific recommendations and suggestions for further research. That is often how I decide if I'm going to read the entire article. With that in mind, I would therefore remove the heading of 4. Limitations and Strengths and place lines 274-290 after the paragraph ending at line 250. I suggest adding a 5. Conclusion section that consists of lines 251-272, with just a minimal rewrite of line 251. (That is where you find the meat of future issues for research and the importance of treatment intervention.)
Overall, a strong contribution to the literature!
Author Response
Ref: Healthcare-1860122
Manuscript Title: “Cybervictimization and well-being during the outbreak of COVID-19: The mediating role of depression.”
We also want to offer our sincerest thanks to the Reviewers and the Editor for taking the necessary time and effort to review the previous version of the manuscript. We sincerely appreciate all your constructive and thoughtful feedback and suggestions.
In the attached revised manuscript, we have made revisions in response to the concerns that were raised. All changes in the manuscript have been put in red font text. In addition, we have made note of how we replied to each editorial feedback in this document.
Reviewer#1
Comments to the Author
This is a clear, succinct, and well-written article. An important quality of this work is that it is accessible to readers who do not have a background in research methodology/statistics. It is not a surprise that depression is related to cyber-bullying and this article successfully promotes further research attention to adolescent cyber-bullying victimization and its consequences.
A strength of the article is the clear definitions of terms including cybervictimization, well-being, and stressful life events. With 77 references, the literature review is strong.
A sample size of 580 is very good, although I'm sure we'd all prefer for more than 17% of the sample to be male. The authors can make a higher percentage of adolescent males a priority in their follow-up research since males are more likely to be victims of cyberbullying.
While the placement of the paragraphs in the sections of 4. Discussion and 5. Limitations and strengths is okay, I'm going to suggest a bit of a reordering for clarity. When I am skimming a research article, after I look at the abstract, and the introduction, I look to the conclusion--I'm especially looking for specific recommendations and suggestions for further research. That is often how I decide if I'm going to read the entire article. With that in mind, I would therefore remove the heading of 4. Limitations and Strengths and place lines 274-290 after the paragraph ending at line 250. I suggest adding a 5. Conclusion section that consists of lines 251-272, with just a minimal rewrite of line 251. (That is where you find the meat of future issues for research and the importance of treatment intervention.)
Overall, a strong contribution to the literature!
(Response)
Many thanks for the positive comments and suggestions. We'll plan to recruit an enough sample of males in our follow-up research to verify if effetely they are more likely to be victims.on
Now discussion and limits have been modified as suggested.

Reviewer 2 Report
The study seems interesting and genuine , however the authors may need to address the following comments to improve the quality of the manuscript:
- The existing research gap and study objectives should accentuated in the introduction section.
- Although this study may introduce a new evidence, but the authors should clarify why this study is different and why its outcome may introduce new information to the current literature.
- The authors should describe the study tool (the questionnaire), sections, validation process, pilot study if present.
- The authors should also describe the data collection methodology and approach.
- The authors should outline the inclusion and exclusion criteria.
- It is advisable to use passive voice for scientific writing.
- Limitations and conclusions should be written in separate sections and conclusions may be summarized in bullets for clarity.
Author Response
We also want to offer our sincerest thanks to the Reviewers and the Editor for taking the necessary time and effort to review the previous version of the manuscript. We sincerely appreciate all your constructive and thoughtful feedback and suggestions.
In the attached revised manuscript, we have made revisions in response to the concerns that were raised. All changes in the manuscript have been put in red font text. In addition, we have made note of how we replied to each editorial feedback in this document.
Reviewer#2
Comments to the Author
The study seems interesting and genuine, however the authors may need to address the following comments to improve the quality of the manuscript:
(Query 1)
- The existing research gap and study objectives should accentuated in the introduction section.
(Response)
Thanks for the suggestions! Now we have pointed out the gap in the existing research and the study objectives in order to make it more exhaustive.
(Query 2)
- Although this study may introduce a new evidence, but the authors should clarify why this study is different and why its outcome may introduce new information to the current literature.
(Response)
We added some parts to clarify as this study could add new information in the field of cybervictimization.
(Query 3)
- The authors should describe the study tool (the questionnaire), sections, validation process, pilot study if present.
(Response)
We better described the questionnaire considering all the sections. As the instruments used were already validated, we underlined this in the measures section. Therefore, the validation process as well as the pilot study isn't discussed.
(Query 4)
- The authors should also describe the data collection methodology and approach.
(Response)
We improved the methodology section describing the data collection.
(Query 5)
- The authors should outline the inclusion and exclusion criteria.
(Response)
The inclusion/exclusion criteria for data collection were added.
(Query 6)
- It is advisable to use passive voice for scientific writing.
(Response)
We double-checked the text and made some changes.
(Query 7)
- Limitations and conclusions should be written in separate sections and conclusions may be summarized in bullets for clarity.
(Response)
We separated Limitations and Conclusions sections and summarized conclusion in bullets.
